# Antibody Tests in Detecting SARS-CoV-2 Infection: A Meta-Analysis

**DOI:** 10.3390/diagnostics10050319

**Published:** 2020-05-19

**Authors:** Panagiota I. Kontou, Georgia G. Braliou, Niki L. Dimou, Georgios Nikolopoulos, Pantelis G. Bagos

**Affiliations:** 1Department of Computer Science and Biomedical Informatics, University of Thessaly, Papasiopoulou 2-4, 35131 Lamia, Greece; pankontou@gmail.com (P.I.K.); gbraliou@gmail.com (G.G.B.); 2International Agency for Research on Cancer, 69372 Lyon, France; nikidimou@gmail.com; 3Medical School, University of Cyprus, 1678 Nicosia, Cyprus; gknikolopoulos@gmail.com

**Keywords:** antibody test, SARS-CoV-2, IgM, IgG, COVID-19, ELISA

## Abstract

The emergence of Coronavirus disease 2019 (COVID-19) caused by SARS-CoV-2 made imperative the need for diagnostic tests that can identify the infection. Although Nucleic Acid Test (NAT) is considered to be the gold standard, serological tests based on antibodies could be very helpful. However, individual studies are usually inconclusive, thus, a comparison of different tests is needed. We performed a systematic review and meta-analysis in PubMed, medRxiv and bioRxiv. We used the bivariate method for meta-analysis of diagnostic tests pooling sensitivities and specificities. We evaluated IgM and IgG tests based on Enzyme-linked immunosorbent assay (ELISA), Chemiluminescence Enzyme Immunoassays (CLIA), Fluorescence Immunoassays (FIA), and the Lateral Flow Immunoassays (LFIA). We identified 38 studies containing data from 7848 individuals. Tests using the S antigen are more sensitive than N antigen-based tests. IgG tests perform better compared to IgM ones and show better sensitivity when the samples were taken longer after the onset of symptoms. Moreover, a combined IgG/IgM test seems to be a better choice in terms of sensitivity than measuring either antibody alone. All methods yield high specificity with some of them (ELISA and LFIA) reaching levels around 99%. ELISA- and CLIA-based methods perform better in terms of sensitivity (90%–94%) followed by LFIA and FIA with sensitivities ranging from 80% to 89%. ELISA tests could be a safer choice at this stage of the pandemic. LFIA tests are more attractive for large seroprevalence studies but show lower sensitivity, and this should be taken into account when designing and performing seroprevalence studies.

## 1. Introduction

In December 2019, a pneumonia outbreak occurred in Wuhan in China due to a new coronavirus that was later officially named SARS-CoV-2 by the World Health Organization (WHO) [1,2]. The disease rapidly spread worldwide, and on February 24, WHO declared COVID-19 (coronavirus disease 2019) a pandemic [3]. SARS-CoV-2 shares pathogenicity features with the human coronaviruses SARS-CoV and MERS-CoV [4], but the incubation period is longer (up to 14 days) [3]. Most patients exhibit mild symptoms, and only a few cases progress to severe or critical disease. Risk factors for severe disease include older age [5] and comorbidities such as hypertension, diabetes, chronic obstructive pulmonary disease (COPD), and cardiovascular disease [6], whereas a higher incidence in males has also been reported [7].

The genome of SARS-CoV-2 is predicted to encode 4 structural proteins (including Spike (S), and Nucleocapsid (N)), 8 accessory, and 15 non-structural proteins [8]. The S protein comprises the receptor binding domain (RBD), which is responsible for binding to the ACE2 membrane receptor of the host cell [9,10,11,12]. The N protein is the structural helical nucleocapsid protein of the virus and is important for transcription and viral replication and packaging [13,14]. The S and N proteins show high antigenicity [15,16,17]. 

Although rigorous public health measures have been taken globally including mass quarantine, COVID-19 incidence is rising leading to 2,402,980 laboratory-confirmed cases and over 165,641 deaths worldwide by April 20. Due to the ongoing COVID 19 outbreak, there is an urgent global need for diagnostic tests. WHO suggests that detection of SARS-CoV-2 nucleic acid (*E* gene followed by the *RdRp* gene) is performed in respiratory samples [18,19,20], while the United States Centers for Disease Control (CDC) recommends the nucleocapsid protein targets N1 and N2 [21]. However, the global shortage of diagnostic tests and especially of swabs for collecting respiratory samples, the frequency of false negative results, and the inability of these tests to be performed in a balk and quick manner that is often required at hospital admission highlight the necessity to develop additional testing methods.

COVID-19 serological tests are mainly based on detecting specific antibodies against SARS-CoV-2 antigens. IgM are the first antibodies that appear in response to the initial exposure to an antigen, while IgG appear later and are more specific to the antigen. COVID-19 serological tests for IgG and IgM have been developed by many laboratories and companies and can be useful in various ways: (a) they can confirm Nucleic Acid Tests (NAT) results or detect infected people who were negative according to NATs [22]; (b) they are cheap, quick, and amenable to rapid broad screening at points of care (POC); (c) blood/serum samples that are used show reduced heterogeneity compared to respiratory specimens; and (d) blood/serum sampling encompasses lower risk for health care workers compared to respiratory sampling where patients are more likely to disperse the virus. Additionally, serological assays can help determine the immune status of individuals [15] and estimate herd immunity. 

Since all the above serological tests have been developed rapidly and under urgent market demands, they are poorly validated with clinical samples in everyday practice. Within several studies, these tests show divergence in sensitivity and specificity that may deviate from what the manufacturers report. Given the importance of serological tests in combating COVID-19, this systematic review and meta-analysis aims to summarize the available evidence on the performance of all available antibody-tests for SARS-CoV-2. 

## 2. Materials and Methods 

### 2.1. Search Strategy and Selection Criteria

For conducting the systematic review and the meta-analysis we followed the Preferred Reporting Items for Systematic reviews and Meta-analyses (PRISMA) guidelines [23] and the advises for best practices [24]. We conducted the literature search using PubMed (https://www.ncbi.nlm.nih.gov/pubmed/), medRxiv (https://medrxiv.org/) and bioRxiv (https://www.biorxiv.org/). The search terms used were (SARS-CoV-2 OR “Coronavirus disease 2019” OR COVID-19) AND (IgM OR IgG or antibodies OR antibody OR ELISA or “rapid test”). The references of selected articles were also searched. The searches were concluded by April 17, 2020, and four different researchers independently evaluated search results. Disagreements in the initial evaluation were resolved by consensus. We did not impose language criteria and included studies written in English and Chinese. We required that eligible studies met the following criteria: (a) COVID-19 cases (SARS-CoV-2 infection) were confirmed either by NAT such as RT-PCR or sequencing or by a combination of NAT and clinical findings and (b) measurements of IgM and/or IgG antibodies were obtained with the use of any of the available methods. We considered eligible studies reporting the comparison of COVID-19 cases against non COVID-19 individuals, as well as case series reporting data only from COVID-19 patients.

Data extracted for each study included (if available): first author’s last name, percentage of male patients, mean age of COVID-19 patients, mean number of days from onset, and percentage of severe or critically ill COVID-19 patients. In addition, the different bioanalytical methods used for detection and determination of IgG and IgM were also recorded, along with the antigen used to detect the antibodies. In order to construct the 2 × 2 contingency table and obtain estimates for sensitivity and specificity, we recorded the numbers of true positive (TP), false positive (FP), true negative (TN), and false negative (FN) for each study. For studies reporting only COVID-19 patients, we recorded only TP and FN. 

The immunoassay methods used for COVID-19 antibody (Ab) detection in all studies included in the present meta-analysis comprise enzyme-linked immunosorbent assay (ELISA), chemiluminescence immunoassays (CLIA), fluorescence immunoassays (FIA), and the point-of-care (POC) lateral flow immunoassays (LFIA) that are based on immunochromatography [25,26,27,28,29]. All methods were created to detect IgG and/or IgM antibodies (and in few cases total antibodies) [30,31,32] against S (mainly RBD) and/or N viral proteins of human sera/blood samples. For detection of IgM with ELISA, the μ-chain capture principle was used. Plates were firstly coated with mouse anti-human IgM (μ chain) monoclonal antibody. Diluted serum samples (heat-inactivated) together with positive and negative controls were added into the pre-coated plates according to individuals’ protocols and incubated at 37 °C for usually 1 h. Washing followed, and horse radish peroxidase (HRP) conjugated recombinant protein of SARS-CoV-2 (rN or rS produced in-house or obtained from a company) was added. After incubation of plates at 37 °C and washes, Tetramethylbenzidine (TMB) substrate solution and the corresponding buffer were added followed by incubation at 37 °C. The reactions were terminated by sulfuric acid addition and the absorbance values at 450 nm (A_450_) were determined. Usually, the cut off values were calculated by sum and average A_450_ of negative control replicates. The principle for the detection of IgG was indirect ELISA, where serum sample dilutions (plus positive and negative controls) were added to previously coated with rN or rS protein ELISA plates. Subsequently, incubation and washes were performed, and HRP-conjugated mouse anti-human IgG monoclonal antibody was added into the plates. Absorbance values at 450 nm were measured for detection. The cut off value was calculated by the sum and average A_450_ of negative control replicates [27]. 

LFIA is a rapid method based on immunochromatography, which uses colloidal gold conjugated COVID-19 antigens. It comprises a plastic pad where a nitrocellulose membrane is fitted. Three separate lines are created by immobilizing goat anti-human IgM, IgG, and goat anti-rabbit-IgG at test M, G, and control (C) lines, respectively. The entire conjugate pad is sprayed with a mixture of AuNP-COVID-19 recombinant antigen conjugate (colloidal-gold pretreated with SARS-CoV-2 recombinant protein) and AuNP-rabbit-IgG. Sample is applied to the sample pad and with the aid of a buffer migrates towards the immobilized lines of antibodies spread with the AuNP-recombinant antigen. When a reaction occurs, a visible line is formed suggesting the existence of IgM and IgG. It should be noted that color in the control line should be formed for a test to be valid [26,33]. The test gives qualitative results that are judged by optical inspection, usually 15 min after sample application. In some LFIAs purchased from companies, the specific antigen that LFIA was based on was not reported. Since most of the companies provide combined N- and S-based LFIAs, we assumed that in unspecified cases the LFIAs were N- and S-based. 

CLIA is a chemiluminescence-based assay, mainly developed by companies. The detection of IgG or IgM is based on double-antibodies sandwich immunoassay. Recombinant antigens rN and rS are conjugated with fluorescein isothiocyanate (FITC) and immobilized on the anti-FITC antibody conjugated magnetic particles. Alkaline phosphatase conjugated human IgG/IgM antibody is used as the detection antibody. An automated magnetic chemiluminescence analyzer is needed to read the measured values of chemiluminescence, and results are given as arbitrary units. As threshold, 10 AU/mL is usually used for both IgM and IgG and according to manufacturer recommendations [34]. The analyzer can be batch and random access with the possibility to give results within half an hour at best [34,35]. Because in most cases CLIA detected both anti-N and anti-S IgG and IgM antibodies (with only one study detecting anti-N [34]), we assumed N- and S-based IgG and IgM CLIAs in studies without relevant information.

With FIA, we denote fluorescence immunoassays that can be performed on multi-test cover slides [36] or be based on fluorescence immunochromatography (AIE/Quantum dot-based fluorescence immunochromatographic assay, AFIA) [37,38]. The latter can be rapid, but all fluorescence-based immunoassays need analyzers to read the results [38]. 

### 2.2. Data Analysis

We performed a quality assessment of the included studies using the Quality Assessment of Diagnostic Accuracy Studies 2 (QUADAS-2) tool, offered by the Review Manager Software (RevMan 5.2.3) (Appendix A). The QUADAS is a quality assessment tool specifically developed for systematic reviews of diagnostic accuracy studies and consists of four key domains: patient selection, index test, reference standard, and flow and timing; each domain is rated as low risk, high risk and unclear risk (Appendix A).

We used the bivariate meta-analytic method modified for the meta-analysis of diagnostic tests [39]. The method has been shown to be equivalent to the so-called hsROC method [40,41] and uses logit-transforms of TPR (true positive rate) and FPR (false positive rate) in order to model Sensitivity and Specificity, as well as to account for the between-studies variability (heterogeneity). Studies that include information only for logit (TPR) are included under the missing at random assumptions in order to maximize the sample and allow for modelling the between-studies variability and correlation. The Begg’s rank correlation test [42] and the Egger’s regression test [43] were used on logit (TPR) to evaluate possible publication bias. The analysis was performed using Stata 13 (Stata Corporation, College Station, TX, USA) and the command “mvmeta” with the method of moments for multivariate meta-analysis and meta-regression [44]. Statistical significance was set at *p* < 0.05. Meta-analysis was performed in cases where two or more studies were available whereas meta-regression and tests for publication bias where 5 or more studies were available. 

## 3. Results

The electronic search revealed 115 articles from PubMed, 72 from medRxiv and 12 from bioRxiv, from which we identified 38 eligible studies after scrutiny [25,26,27,28,29,30,31,32,33,34,35,36,37,38,45,46,47,48,49,50,51,52,53,54,55,56,57,58,59,60,61,62,63,64,65,66,67,68] (Figure 1). These include in total 7848 individuals (3522 COVID-19 cases and 4326 healthy, or non COVID-19, individuals). A total of 21 studies reported data for both COVID-19 cases and controls, whereas 17 studies reported data only for COVID-19 cases (Table 1). A total of 13 studies used RT-PCR or other nucleic acid-based tests (NATs) as the gold standard for case ascertainment, whereas 25 studies ascertained COVID-19 cases using a combination of molecular and clinical features. We built our analysis on grouping the tests according to the method and the specific antigen used. Because we found kits and reagents from 25 different companies, plus the various in-house tests developed for research purposes, stratification according to different kits was pointless. Several studies reported the results of multiple tests on the same individuals; however, they were not included in the same meta-analysis since we analyzed each test separately. In one study that compared several different LFIA tests, we used the results of the one with the median performance (even though the differences were small). Other studies reported samples from multiple populations, and in such cases, they were considered distinct. 

14 studies in total reported results from ELISA-based tests (detecting anti-N or anti-S IgG, IgM antibodies, or both). S-based ELISAs, in general, perform better compared to those based on N antigen. IgG and IgM seem to perform similarly, but the combination of IgG and IgM seems to be superior leading to a sensitivity of 0.935 (95% CI: 0.900, 0.971) (Figure 1). All ELISA-based methods seem to have rather high specificities (ranging from 0.961 to 0.995). Meta-regression analysis showed that the mean number of days from disease onset and the proportion of severe/critical patients have an influence on the overall sensitivity of the IgG tests. Both Egger’s and Begg’s tests did not detect publication bias or other small study effects.

CLIA-based tests were used in 13 studies. In all cases IgGs and IgMs were investigated. In this analysis, we also pooled together the studies that considered NS antigens with the studies that used N antigen. The sensitivities of detecting IgG seem to be better compared to that of IgM (0.944 vs. 0.810). Combining IgM and IgG yields a slightly worse sensitivity (0.907, 95% CI: 0.753, 1.000), but this estimate arises from only two studies (970 patients) and thus has large uncertainty (Figure 1). Specificities range from 0.954 to 0.984. Meta-regression analysis revealed that the mean number of days from disease onset has an influence on the overall outcome in the IgG tests. The Begg’s test provided some evidence for publication bias in the IgG analysis.

13 studies reported results from LFIA-based tests. The majority of the tests identified antibodies against both N and S antigens, and results were obtained for both IgG and IgM. In this analysis, we also pooled together the studies that considered NS antigens with the studies that used S antigen. IgG and IgM seem to perform comparably but rather low since the sensitivities range from 0.53 to 0.66. Combining IgG and IgM yields better estimates (0.78–0.83) but still with lower sensitivity compared to ELISA- and CLIA-based tests (Figure 1). Specificity in all cases ranged from 0.914 to 0.994. In the largest overall analysis, pooling together the 11 studies that used N, S, or NS antigens, the combination of IgG and IgM antibodies yields a sensitivity of 0.800 (95% CI: 0.663, 0.935) and specificity of 0.984 (95%CI: 0.969, 0.999). Meta-regression analysis revealed that the mean number of days from disease onset influences the overall outcome in the IgG and IgG/IgM tests. Both Egger’s and Begg’s tests could not find evidence for publication bias or other small study effects.

Lastly, FIA-based tests were found in three studies using a combination of N and S antigens. Both IgG and IgM show similar sensitivities (~0.86) and specificities (0.95) (Figure 2, Figure 3); however, the sample is small (3 studies, 327 patients). Due to the small number of studies, tests for publication bias or meta-regression could not be applied.

## 4. Discussion

Non-pharmaceutical interventions including increased testing rates, contact tracing, school closures, ban of mass gatherings, physical distancing, restriction of movement, and cordon sanitaire were effective in reducing transmission rates of SARS-CoV-2 in Wuhan, China, and other settings [69]. However, this type of intervention has tremendous societal and economic consequences potentially resulting in social disorganization and great recession. One approach to de-escalating public health measures and returning to a state of normalcy, while maintaining epidemiological vigilance and ability to respond fast to viral resurgence, is to identify people with immunity to SARS-CoV-2 and estimate their proportion in the entire population. This approach would indicate immune people including health-care workers who can go back to work without risking their health or that of others, help reopen borders, and monitor the development of herd immunity. Unfortunately, human immune response to the new pathogen is not well studied yet. The serological tests that have recently been developed employ different methods and target either IgG or IgM or both. In an attempt to fill the knowledge gap, this systematic review summarized evidence from 38 studies involving 7848 individuals. Although the US Food and Drug Administration (FDA) has approved ELISA, LFIA, and neutralization assays, we included in the present meta-analysis studies using CLIA and FIA methods as well, because they can potentially be approved in the future. We did not consider neutralization assays since they are more time demanding (3–5 days) and can only be performed in laboratories of Biosafety Level-3 (BSL-3) [70]. The meta-analysis showed that all methods yielded high specificity with some of the methods (ELISA and LFIA) reaching levels higher than 99%. ELISA- and CLIA-based methods performed better in terms of sensitivity (90–96%) followed by LFIA and FIA with sensitivities ranging from 80% to 89%. 

Sample quality, low antibody concentrations, and especially timing of the test—too soon after a person is infected when antibodies have not been developed yet or too late when IgM antibodies have decreased or disappeared—could potentially explain the low ability of the antibody tests to identify people with COVID-19 [70]. According to kinetic measurements of some of the included studies [22,50,55] IgM peaks between days 5 and 12 and then drops slowly. IgGs reach peak concentrations after day 20 or so as IgM antibodies disappear. This meta-analysis showed, through meta-regression, that IgG tests did have better sensitivity when the samples were taken after the first week that follows the onset of symptoms. This is further corroborated by the lower specificity of IgM antibodies compared to IgG [15]. Only few of the included studies provided data stratified by the time of onset of symptoms, so a separate stratified analysis was not feasible, but this should be a goal for future studies. In any case, care should be paid when antibody tests are used in the first week after the onset of disease symptoms. Moreover, irrespective of the method, a combined IgG/IgM test seems to be a better choice in terms of sensitivity than measuring either antibody type alone. The analyses also showed that tests that use the S antigen are more sensitive than N antigen-based ELISA tests, probably due to higher sensitivity and earlier immune response to the S antigen [53] and more specific perhaps due to lower cross-reactivity with less conserved regions of spike proteins existing in other coronaviruses (SARS-CoV) [17,56,64]. Finally, despite the suboptimal sensitivity, antibody tests could certainly supplement NATs in the diagnosis of people with suspected SARS-CoV-2 infection [65]. In any case, a direct comparison of antibody tests against NATs is also needed in future studies (in the current review only a handful of studies performed this, and they did that only in COVID-19 patients).

Antibody tests for SARS-CoV-2 have other accuracy issues that deserve attention and further assessment. For instance, cross-reaction with human endemic coronaviruses could make antibody tests less specific and produce false positive results [30,35,56,63]. A low specificity may have important consequences both in terms of diagnosis and population surveillance. At the individual level, false positive results pose risks as people who have never been infected are perhaps allowed to work or travel because they are considered immune. At a population level and regarding epidemiological studies, given the low prevalence of SARS-CoV-2 in most settings at the moment, false positives may inflate prevalence estimates and give a distorted picture of lower mortality rate and higher population immunity than what is in reality. On the other hand, low sensitivity may result in falsely assuming that a person is not infected and consequently jeopardizing measures to prevent the spread of the epidemic. Based on the results of this meta-analysis, ELISA tests that achieved specificity higher than 99% and sensitivity ~93% could be the safer choice at this stage of the pandemic. CLIA tests show comparable sensitivity (~90%) but slightly decreased specificity (95–98%). LFIA tests on the other hand are particularly attractive for large seroprevalence studies and can be used as POC tests. They show high specificity, comparable to ELISA (~99%), but lower sensitivity (~80%), and these estimates should be taken into account when designing and performing seroprevalence studies, for instance, by adjusting properly the obtained positive and negative findings. At the individual level, perhaps mixed strategies could be adopted (for instance re-testing a negative finding using a different test).

Of note, even if tests are highly accurate, much about protective immunity is unknown, and the true presence of binding antibodies might not mean that people have indeed developed high titers of neutralizing antibodies and are thus immune to re-infection [71]. Research on Rhesus macaques infected with SARS-CoV-2 was promising though showing that reinfection did not occur following rechallenge with the same dose of SARS-CoV-2 strain [72]. Finally, viral load does not decline rapidly after seroconversion and people may remain infectious despite being truly positive in antibodies tests [36].

## Figures and Tables

**Figure 1 diagnostics-10-00319-f001:**
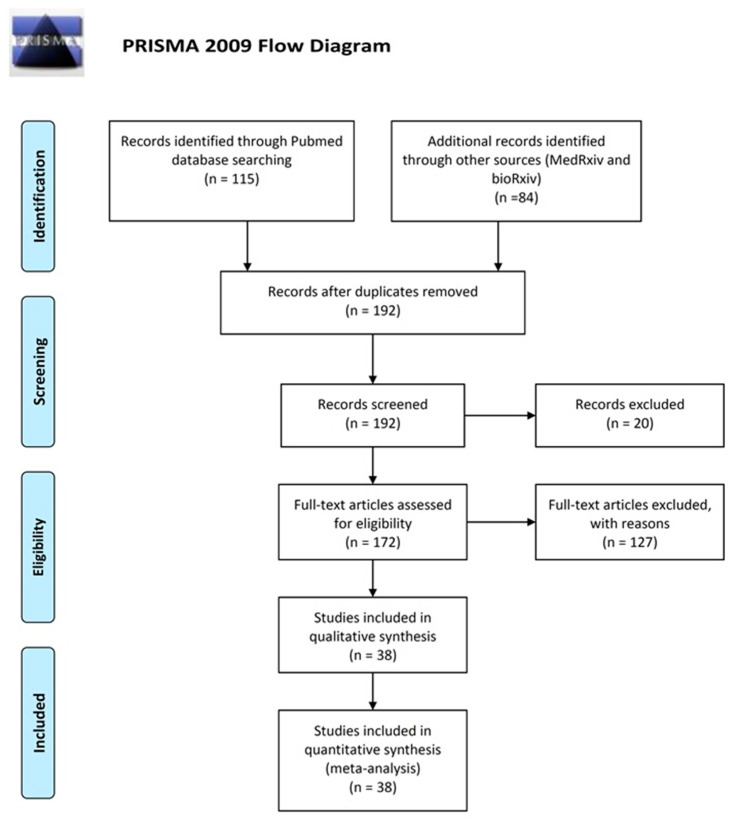
Preferred reporting items for systematic reviews and meta-analyses (PRISMA) flow diagram.

**Figure 2 diagnostics-10-00319-f002:**
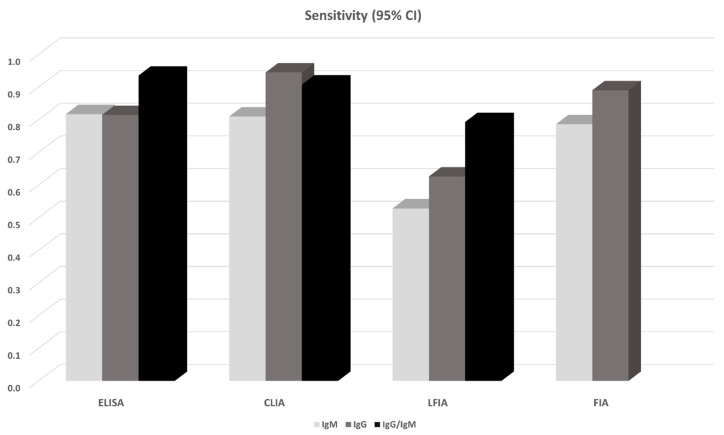
Pooled sensitivity of antibody tests obtained from meta-analysis. For the details see Table 2 and the Results section.

**Figure 3 diagnostics-10-00319-f003:**
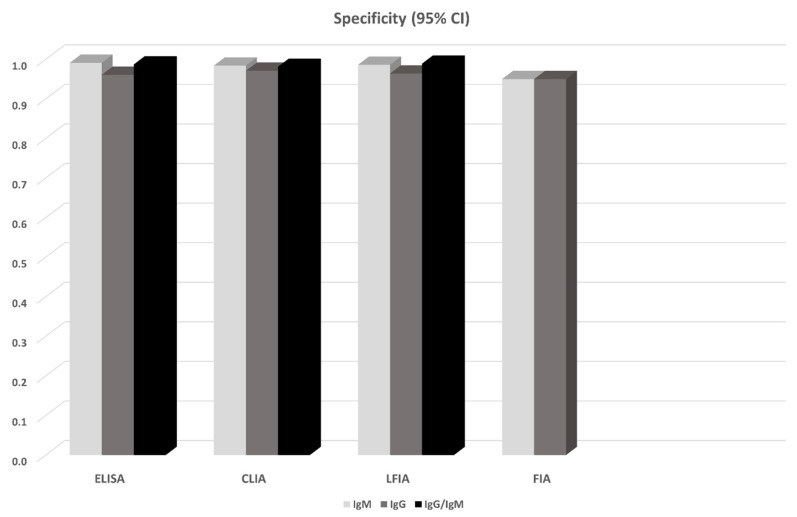
Pooled specificity of antibody tests obtained from meta-analysis. For the details see Table 2 and the Results section.

**Table 1 diagnostics-10-00319-t001:** Characteristics of the 38 studies included in the meta-analysis.

Author [Ref]	Covid19/Healthy	Covid19 Ascertainment	Severe Covid19 (%)	Male Cases (%)	Cases Age	Days from Onset	Antibodies	Method	Company	Limit of Detection IgM/IgG	Sensitivity	Specificity
Liu [27]	238/120	RT-PCR/clinical features	NR	58	55	14	IgM (N)/IgG (N)	ELISA	ZhuHai LivZon, Diagnostics	A_450_: 0.100/0.130	0.11–0.81	0.96–0.99
Long [55]	262/148	RT-PCR	13.6	55.4	47	13	IgM (N,S)/IgG (N,S)	CLIA	Bioscience (Chongqing) Co., Ltd.	NR	0.67–0.80	0.95
Jia [38]	33/242	NR-NAT/clinical features	NR	NR	NR	15	IgM (N,S)/IgG (N,S)	FIA	Beijing Diagreat Biotechnologies Co., Ltd.	Fluorescence Intensity: 0.88/1.02 (Flu units)	0.45–0.72	0.95
Liu [54]	95/84	RT-PCR	49	70	76	18	IgM (N)/IgG (N)	LFIA	Not Reported (a Chinese Company)	NA	0.37–0.86	0.93–0.94
Xu [33]	10/0	NAT/sequencing	100	60	NR	22	IgM (S)/IgG (S)	LFIA	In-house test	NA	0.3–0.9	NA
Wang [34]	116/0	RT-PCR/clinical features	12.9	56	68.8	31	IgM (N,S)/IgG (N,S)	CLIA	YHLO Biotechnology (Shenzhen, China)	10 AU/mL	1	NA
Xiang [28]	63/35 ELISA, 91/35 LFIA	RT-PCR/clinical features	6.3	55.5	57.82	NR	IgM (N,S)/IgG (N,S)	ELISA/LFIA	ZhuHai LivZon, Diagnostics Inc.BioEasy/Shenzhen BioEasy Biotechnology Co.	NR/NA	0.44–0.87	1
Zhang [64]	154/660	RT-PCR/clinical features	NR	NR	NR	NR	IgM (S)/IgG (S)	LFIA	In-house test	NA	0.82	0.99
Lin [35]	79/80	RT-PCR/clinical features	NR	35	43.6	14	IgM (N)/IgG (N)	ELISA/CLIA	Darui Biotech, China/Tianshen Tech, Shenzhen, China	NR/NR	0.23–0.91	0.78–1
Hu [37]	34/9	RT-PCR	NR	NR	NR	NR	IgM (N,S)/IgG (N,S)	FIA	KingFocus Biomedical engineering Co., Ltd.	Cutoff values were based on of seronegative samples	0.97-1	1
Zhang [32]	222/0	RT-PCR	39.2	48.2	64	20	IgM (N,S)/IgG (N,S)	CLIA	YHLO Biotechnology (Shenzhen, China) and the high-speed CLIA system iFlash 3000, BATCH ANALYZER	Cutoff values were based on of seronegative samples	0.83–0.99	NA
Okba [56]	12/0	RT-PCR	NR	NR	NR	11	IgG (S)	ELISA	EUROIMMUN Medizinische Labordiagnostika AG	Cutoff values set by mean of seronegative samples plus 6SD	0.92	1
Zhang [63]	3/733	RT-PCR/clinical features	66.6	66.6	50.67	10	IgM (N,S)/IgG (N,S)	CLIA	YHLO Biotechnology (Shenzhen, China)	10 AU/mL	1	0.98
Zhao [66]	69/412	NR-NAT/clinical features	NR	NR	NR	NR	IgM (S)/IgG (S)	ELISA	In-house test	Cutoff values were based on seronegative samples	0.97	0.97
Pan [57]	86/0	RT-PCR/clinical features	NR	45.7	58	12	IgM (N,S)/IgG (N,S)	LFIA	ZhuHai LivZon, Diagnostics	NA	0.55–0.69	NA
Lou [31]	80/300	RT-PCR/clinical features	33	61.3	55	15	IgM (N,S)/IgG (N,S)	ELISA/CLIA/LFIA	Beijing Wantai Biological Pharmacy Enterprise Co., Ltd., China (Beijing, China)/Xiamen InnoDx Biotech Co., Ltd.	NR/NR/NA	0.86–0.97	0.95–1
Liu [27]	133/0	RT-PCR/clinical features	66.9	52.6	68.5	NR	IgM (N,S)/IgG (N,S)	CLIA	YHLO Biotechnology (Shenzhen, China)	10 AU/mL	0.79–0.97	NA
Tan [59]	65/0	RT-PCR/clinical features	43.3	52.2	49	15	IgM (N)/IgG (N)	ELISA	ZhuHai LivZon, Diagnostics	Titer cutoff value set according to non-responders	0.43–0.78	NA
To [60]	16/0	RT-PCR/sequencing/clinical features	43.5	56.5	62	20	IgM (N,S)/IgG (N,S)	ELISA	In-house test	Cutoff set by mean of seronegative samples plus 3SD	0.87–1	NA
Xiao [29]	34/0	RT-PCR/clinical features	NR	64.7	55	25	IgM (N,S)/IgG (N,S)	CLIA	YHLO Biotechnology (Shenzhen, China)	10 AU/mL	0.82–0.94	NA
Cassaniti [47]	30/38	RT-PCR	NR	83.3	73.5/61.5	7	IgM (N,S)/IgG (N,S)	LFIA	VivaChekTM	NA	0.13–0.83	1
Liu [53]	214/100	RT-PCR	NR	NR	NR	15	IgM (N,S)/IgG (N,S)	ELISA	ZhuHai LivZon, Diagnostics	A_450_: 0.100/0.130	0.68–0.77	1
Li [26]	397/128	RT-PCR	NR	NR	NR	20	IgM (S)/IgG (S)	LFIA	Jiangsu Medomics Medical Technologies	NA	0.7–0.82	0.91
Zhao [65]	173/0	RT-PCR/clinical features	18.5	48.5	48	7	IgM (S)/IgG (S)	ELISA	Beijing Wantai Biological Pharmacy Enterprise Co., Ltd.	Cutoff value set by seronegative samples	0.65–0.93	NA
Bai [45]	6/0	RT-PCR/clinical features	16.7	50	49	2	IgM (N,S)	LFIA	Institute of Microbiology and Epidemiology of the Military Medical Research Institute and Beijingh Rejing Biotecnology Co., Ltd.	NA	0.83	NA
Zheng [67]	55/0	RT-PCR/clinical features	40	43.6	60	11	IgM (N,S)/IgG (N,S)	CLIA	Not Reported	NR	0.82–0.98	NA
Zeng [61]	6/0	RT-PCR/clinical features	0	0	NR	NR	IgM (N,S)/IgG (N,S)	CLIA	YHLO Biotechnology (Shenzhen, China)	10 AU/mL	0.83	1
Guo [50]	140/285	RT-PCR/sequencing/clinical features	23.6	NR	NR	13	IgM (N)	ELISA	In-house test	A_450_: 0.130/0.300	0.83	1
Jin [51]	27/33	RT-PCR	0	39.5	47	16	IgM (N,S)/IgG (N,S)	CLIA	YHLO Biotechnology (Shenzhen, China)	10 AU/mL	0.48–0.89	0.9-1
Du [25]	60/0	NR-NAT/clinical features	NR	NR	NR	43	IgM (N,S)/IgG (N,S)	CLIA	YHLO Biotechnology (Shenzhen, China)	10 AU/mL	0.78–1	NA
Wölfel [36]	9/0	RT-PCR/clinical features	0	NR	NR	18	IgM (S)/IgG (S)	FIA	In-house with reagents from Euroimmun AG, Lübeck, Germany	NR	0.66–1	NA
Zhong [68]	47/300	NR-NAT	23.4	34	48.21	15	IgM (N,S)/IgG (N,S)	ELISA / CLIA	In-house test	A_450_: IgM(N) 0.059, IgM(S) 0.167/IgG(N) 0.036, IgG(S) 0.079/NR	0.89–0.98	0.85–0.97
Lassaunière [30]	30/82	RT-PCR	100	NR	NR	NR	IgM (N,S)/IgG (N,S)	ELISA / LFIA	Εuroimmun Medizinische Labordiagnostika, Lübeck, Germany/Beijing Wantai Biological Pharmacy Enterprise, Beijing, China/Dynamiker Biotechnology, Tianjin, China/CTK Biotech, Poway, CA, USA/AutoBio Diagnostics, Zhengzhou, China/Artron, Laboratories, Burnaby, Canada	NR/NA	0.66–0.93	0.95–1
Gao [48]	38/0	RT-PCR/clinical features	7.9	55.3	40.5	16	IgM (N,S)/IgG (N,S)	LFIA	Innovita Biological Technology Co., Ltd.	NA	0.51–0.92	NA
Zeng [62]	27/36	RT-PCR/clinical features	63	51.8	62	18	IgM (N)/IgG (N)	ELISA	ZhuHai LivZon, Diagnostics	A_450_: 0.105/0.105	1	1
Garcia [49]	118/45	RT-PCR/clinical features	NR	67.8	65.14	14	IgM (N,S)/IgG (N,S)	LFIA	Biotech AllTest, Hangzhou, China	NA	0.31–0.69	1
Paradiso [58]	191/0	RT-PCR/clinical features	NR	60.62	58.5	4	IgM (N,S)/IgG (N,S)	LFIA	VivaChekTM	NA	0.14–0.16	NA
Bendavid [46]	122/456	RT-PCR	NR	NR	NR	NR	IgM (N,S)/IgG (N,S)	LFIA	Premier Biotech	NA	0.67–0.92	0.99–1

Severe Covid19 (%): Percentage of severe cases with Covid19. Male cases (%): Percentage of male cases. NR: not reported. NR-NAT: A nucleic acid test was used but the exact type of the test was not reported. NA: Not applicable. Multiple values for specificity and specificity are recorded in each study because different assays were used in most cases (i.e., IgG, IgM, and so on).

**Table 2 diagnostics-10-00319-t002:** Results of the meta-analysis for the different types of antibody tests. We list the characteristics of the included studies, the pooled sensitivity and specificity along with the 95% confidence intervals, the variables that were found statistically significant in meta-regression, and the results of the tests for publication bias. For the description of the test, the antibodies (Ab) and antigens (Ag), see Methods section (mdfo: mean days from onset; severe: percent of patients with severe or critical condition; NA: not applicable). N: nucleocapsid protein, S: spike protein, NS: nucleocapsid and Spike proteins.

Method	Ab	Ag	Studies/Patients	Sensitivity (95% CI)	Specificity (95% CI)	Covariates	Begg’s/Egger’s
ELISA	IgG	N	8/1472	0.747 (0.509, 0.984)	0.994 (0.988, 0.999)	mdfo, severe	-/-
ELISA	IgG	S	7/1072	0.814 (0.688, 0.940)	0.961 (0.910, 1.000)	-	-/-
ELISA	IgM	N	8/1717	0.722 (0.449, 0.996)	0.995 (0.989, 1.000)	-	-/-
ELISA	IgM	S	6/1328	0.817 (0.704, 0.931)	0.991 (0.976, 1.000)	-	-/-
ELISA	IgG/IgM	N	2/423	0.808 (0.764, 0.853)	0.967 (0.915, 0.987)	NA	NA
ELISA	IgG/IgM	S	5/1244	0.935 (0.900, 0.971)	0.987 (0.973, 1.000)	-	-/-
LFIA	IgG	S	2/535	0.537 (0.123, 0.951)	0.914 (0.853, 0.951)	NA	NA
LFIA	IgG	NS	8/944	0.650 (0.404, 0.895)	0.988 (0.973, 1.000)	mdfo	-/-
LFIA	IgG	S/NS	10/1479	0.626 (0.439, 0.814)	0.964 (0.922, 1.000)	-	-/-
LFIA	IgM	S	2/535	0.663 (0.236, 1.000)	0.914 (0.852, 0.951)	NA	NA
LFIA	IgM	NS	9/1059	0.528 (0.329, 0.726)	0.986 (0 974, 0.998)	-	-/-
LFIA	IgM	S/NS	11/1594	0.555 (0.352, 0.758)	0.979 (0.958, 0.999)	-	-/-
LFIA	IgG/IgM	S	2/824	0.828 (0.770, 0.886)	0.994 (0.984, 0.998)	NA	NA
LFIA	IgG/IgM	NS	8/1373	0.777 (0.592. 0.962)	0.986 (0.973, 1.000)	mdfo	-/-
LFIA	IgG/IgM	S/NS	10/2197	0.793 (0.643, 0.942)	0.989 (0.978, 0.999)	mdfo	-/-
LFIA	IgG/IgM	S/N/NS	11/2376	0.800 (0.663, 0.935)	0.984 (0.969, 0.999)	mdfo	-/-
CLIA	IgG	NS	12/2320	0.944 (0.906, 0.983)	0 971 (0.931, 1.000)	mdfo	-/+
CLIA	IgG	N/NS	13/2479	0.935 (0.896, 0.975)	0.974 (0.953, 0.994)	mdfo	-/+
CLIA	IgM	NS	12/2411	0.810 (0.722, 0.897)	0.984 (0.970, 0.999)	-	-/-
CLIA	IgM	N/NS	13/2570	0.799 (0.737, 0.860)	0.967 (0.927, 1.000)	-	-/-
CLIA	IgG/IgM	NS	2/790	0.907 (0.753, 1.000)	0.981 (0.944, 1.000)	NA	NA
CLIA	IgG/IgM	N/NS	3/949	0.902 (0.811, 0.993)	0.954 (0.875, 1.000)	NA	NA
FIA	IgG	NS	2/318	0.859 (0.339, 1.000)	0.950 (0.923, 0.977)	NA	NA
FIA	IgG	S/NS	3/327	0.890 (0.591, 1.000)	0.950 (0.923, 0.977)	NA	NA
FIA	IgM	NS	2/318	0.860 (0.500, 1.000)	0.950 (0.923, 0.977)	NA	NA
FIA	IgM	S/NS	3/327	0.786 (0.531, 1.000)	0.950 (0.923, 0.977)	NA	NA

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
