# Peer review of "Antibody Tests in Detecting SARS-CoV-2 Infection: A Meta-Analysis"

_diagnostics, 2020, doi:10.3390/diagnostics10050319_

Round 1

Reviewer 1 Report

The review submitted by Kontou and coworkers consisted in a systematic revision of the current state of the art concerning the serological test that have been tested up to now for SARS-cov2. The review has been systematically performed and meta-analysis has been also done. The review is on current interest for the scientific community. In my opinion, the following review is suitable for publication in Diagnostics, however I hope that the authors can consider my comments to improve the quality of the article and clarify some aspects that have been discussed.

During the review article the authors mentioned the Nucleic Acid Test (NAT) as the gold standard to validate the serological analysis. I would suggest to mention as gold standard the RT-PCR, although other test based on RNA can be considered to validated. WHO and FDA describe in their guidelines the use of PCR as reference methods. If there is other, indicate in as footnotes in the table.

Page 3 Line 103-120. I suggest to include reviews that supports the explanation of the basic immunoassays like ELISA, CLIA, FIA or LFIA. Moreover a figure describing all the assays can be helpful for the reader.

Both figures 2 and 3 represent the sensitivity and specificity performance of the antibody test. For one side, the legend has to be rewritten propertly with the aim that the figures can be auto-explicative. For other side, 95%CI doesn’t reflect the variability that can exist between assays. All the antibody assays reviewed perform different, so, I propose to visualize the bars as mean value and SD.

For the Table 1, I suggest to eliminate DOI column. Description in the Descripition table the meaning of the different headings Include Ref in Author heading as Author [Ref]. Meaning of Severe Covid19 (%). Not include decimals I Days from Onset. Indicate the empty cells as not reported or not defined.

From the analytical point of view, I think it is interesting to provide the LOD or LOQ for the different antibody tests in Table 1.

Author Response

The review submitted by Kontou and coworkers consisted in a systematic revision of the current state of the art concerning the serological test that have been tested up to now for SARS-cov2. The review has been systematically performed and meta-analysis has been also done. The review is on current interest for the scientific community. In my opinion, the following review is suitable for publication in Diagnostics, however I hope that the authors can consider my comments to improve the quality of the article and clarify some aspects that have been discussed.

Response:We would like to thank the reviewer for the positive comments. In the revised version of the manuscript we tried to address all the comments and clarify the aspects mentioned.

During the review article the authors mentioned the Nucleic Acid Test (NAT) as the gold standard to validate the serological analysis. I would suggest to mention as gold standard the RT-PCR, although other test based on RNA can be considered to validated. WHO and FDA describe in their guidelines the use of PCR as reference methods. If there is other, indicate in as footnotes in the table.

Response:We changed the column “Covid19 Ascertainment” in table 1 according to the reviewer's suggestion and replaced NAT with RT-PCR and sequencing where appropriate. For studies where no specific method was reported for the determination/quantification of viral nucleic acid we denote NR-NAT (Not reported-Nucleic Acid Test)

Page 3 Line 103-120. I suggest to include reviews that supports the explanation of the basic immunoassays like ELISA, CLIA, FIA or LFIA. Moreover a figure describing all the assays can be helpful for the reader.

Response: According to the reviewer’s instruction we added the explanation of the four immunoassays we refer to in the present meta-analysis. As suggested we incorporated them in the “Materials and Methods” section

Both figures 2 and 3 represent the sensitivity and specificity performance of the antibody test. For one side, the legend has to be rewritten propertly with the aim that the figures can be auto-explicative. For other side, 95%CI doesn’t reflect the variability that can exist between assays. All the antibody assays reviewed perform different, so, I propose to visualize the bars as mean value and SD.

Response:Figure 2 and 3 depict the pooled sensitivities and specificities of the various tests included in the meta-analysis. They show the same numbers with the ones presented in Table 2, but we chose to include them in a graph for reasons of comparison. The reviewer however refers to the sensitivity and specificity of the individual tests, and this is not easily added to such a graph. In order to show this information, we chose to include these figures in Table 1 (columns “sensitivity” and “specificity”)

For the Table 1, I suggest to eliminate DOI column. Description in the Descripition table the meaning of the different headings Include Ref in Author heading as Author [Ref]. Meaning of Severe Covid19 (%). Not include decimals I Days from Onset. Indicate the empty cells as not reported or not defined.

Response: In the revised version we removed the DOI column, we included the Ref in Author heading, we explain the meaning of Severe Covid19 (%), we indicate the empty cells as not reported (NR) and we do not include decimals in Days from Onset as suggested by the reviewer. Moreover, we added the company name of the antibody test used in each included study.

From the analytical point of view, I think it is interesting to provide the LOD or LOQ for the different antibody tests in Table 1.

Response: From the Analytical point of view, precise Ab concentrations are used in cases of pure Abs where the Abs are the sensors of a bioassay.However, in the present immunoassays, Abs are the analyte and due to the urgent character of the diagnosis no quantification efforts have been made by any researcher, by now, according to our knowledge. The approach followed by researchers in this setting involves  cutoff values that are set by averaging the measurements of negative samples from viral nucleic acid negative sera. We added this information in an extra column in table 1 “Limit of Detection IgM/IgG”.

For ELISA method values denote A450 (absorbance at 450 nm) of averaged negative samples, IgM first and IgG second. Similarly, fluorescence arbitrary units are used for the FIA method and the cutoff values are set by the investigators with the use of their negative samples. Values are depicted where information was available.

Concerning the CLIA method, and since a chemilluminesence analyzer is needed, companies have set the threshold cutoff values for both IgG and IgM to 10AU/ml (arbitrary units). This threshold is adopted by all the studies reporting a threshold cutoff included in the present meta-analysis, and is reported in the respective column.

The LFIAs are qualitative immunoassays. The detection limit has not been determined yet by researchers (#24) (doi: 10.1002/jmv.25727) and companies that provide such kits give no information concerning detection limits. Thus, LOD/LOQ information is not applicable (NA) in LFIA.

Reviewer 2 Report

1. As the authors pointed it out the goal of this study is to summarize the available evidence on the performance of all available antibody-tests for SARS-CoV-2. However, the results had explicitly described the specificity and sensitivity of three serological method formats, i.e. ELISA, CLIA, and FLIA, using meta-analysis software. The authors did not go deep to the methodologies to explain possible reason why the sensitivity of these three methods was different, but suggest that the ELISA method may be more preferable. Personally, I don't think meta-analysis can validate the conclusions as the authors suggested. This may cause bias and mis-lead readers to prefer one method over another.

2. Suggest authors to introduce some fundamentals of these three serological methods and some background about using IgG and IgM. In ELISA testing, one could use antibody to detect antigen S or antigen N or use antigen S/N to detect IgG/IgM. Authors did not clearly describe what kind of ELISA formats were included in the study.

Author Response

  1. As the authors pointed it out the goal of this study is to summarize the available evidence on the performance of all available antibody-tests for SARS-CoV-2. However, the results had explicitly described the specificity and sensitivity of three serological method formats, i.e. ELISA, CLIA, and FLIA, using meta-analysis software. The authors did not go deep to the methodologies to explain possible reason why the sensitivity of these three methods was different, but suggest that the ELISA method may be more preferable. Personally, I don't think meta-analysis can validate the conclusions as the authors suggested. This may cause bias and mis-lead readers to prefer one method over another.

Response: We would agree with the above comment of the reviewer that the various immunoassays rely on different methodologies with inherent attributes and variabilities especially in their detection limit. In the revised manuscript we tried to explain these differences. Considering the emergent need of such tests, all these tests/kits came out to the market without having undoubtedly evaluated LoD and LoQ. This is also highlighted in the response to reviewer #1 and in the amendment to Table 1. In any case,  comparing the methods on an analytical basis is beyond the scope of the present manuscript which concentrates on the statistical properties of the diagnostic tests. Given that there are so many different kits from different companies, it would not be feasible to compare them one by one. Thus, grouping the methods and performing meta-analysis by type of the test was the only practical solution at this point. We believe that this approach provides some basic insight on the usefulness of each type of test and a starting point for downstream more detailed analyses when available data emerge.

  1. Suggest authors to introduce some fundamentals of these three serological methods and some background about using IgG and IgM. In ELISA testing, one could use antibody to detect antigen S or antigen N or use antigen S/N to detect IgG/IgM. Authors did not clearly describe what kind of ELISA formats were included in the study.

Response: According to the reviewer’s instruction (as well as the comment of reviewer #1), in the revised manuscript we added the explanation of the four immunoassays that we assessed in the present meta-analysis. As suggested we incorporated them in the “Materials and Methods” section. In addition, in line 61 we added the phrase “COVID-19 serological tests are mainly based on detecting specific antibodies against SARS-CoV-2 antigens. IgM are the first antibodies that appear in response to the initial exposure to an antigen, while IgG appear later, are more specific to the antigen and can build up an individual’s immunity” in order to give some background about using IgG and IgM.  Indeed, the various methods included in the meta-analysis were based on different combinations of antigens (N, S, or NS) and antibodies (IgG, IgM, or both). We think that Table 2 is informative in this respect. Moreover, in various places in the Results and in the Discussion sections we tried to explain the results better.